# Live-Cell Imaging of Single Neurotrophin Receptor Molecules on Human Neurons in Alzheimer’s Disease

**DOI:** 10.3390/ijms222413260

**Published:** 2021-12-09

**Authors:** Klaudia Barabás, Julianna Kobolák, Soma Godó, Tamás Kovács, Dávid Ernszt, Miklós Kecskés, Csaba Varga, Tibor Z. Jánosi, Takahiro Fujiwara, Akihiro Kusumi, Annamária Téglási, András Dinnyés, István M. Ábrahám

**Affiliations:** 1NAP Molecular Neuroendocrinology Research Group, Institute of Physiology, Medical School, Centre for Neuroscience, Szentágothai Research Institute, University of Pécs, 7624 Pécs, Hungary; klaudia.barabas@aok.pte.hu (K.B.); soma.godo@aok.pte.hu (S.G.); tamas.kovacs@aok.pte.hu (T.K.); ernszt.david@pte.hu (D.E.); janosi.tibor@pte.hu (T.Z.J.); istvan.abraham@aok.pte.hu (I.M.Á.); 2BioTalentum Ltd., 2100 Gödöllő, Hungary; biotalentum.julianna@gmail.com (J.K.); annamaria.teglasi@biotalentum.hu (A.T.); 3NAP-B Cortical Microcircuits Research Group, Institute of Physiology, Medical School, Centre for Neuroscience, Szentágothai Research Institute, 7624 Pécs, Hungary; kecskes.miklos@pte.hu (M.K.); csaba.varga@aok.pte.hu (C.V.); 4Institute for Integrated Cell-Material Sciences (WPI-iCeMS), Kyoto University, Kyoto 606-8501, Japan; tfujiwara@icems.kyoto-u.ac.jp; 5Membrane Cooperativity Unit, Okinawa Institute of Science and Technology Graduate University (OIST), Onna 904-0495, Japan; akihiro.kusumi@oist.jp

**Keywords:** Alzheimer’s disease, TrkA, p75^NTR^, receptor dynamics, live-cell single-molecule imaging, neuronal, human-induced pluripotent stem cell

## Abstract

Neurotrophin receptors such as the tropomyosin receptor kinase A receptor (TrkA) and the low-affinity binding p75 neurotrophin receptor p75^NTR^ play a critical role in neuronal survival and their functions are altered in Alzheimer’s disease (AD). Changes in the dynamics of receptors on the plasma membrane are essential to receptor function. However, whether receptor dynamics are affected in different pathophysiological conditions is unexplored. Using live-cell single-molecule imaging, we examined the surface trafficking of TrkA and p75^NTR^ molecules on live neurons that were derived from human-induced pluripotent stem cells (hiPSCs) of *presenilin 1 (PSEN1)* mutant familial AD (fAD) patients and non-demented control subjects. Our results show that the surface movement of TrkA and p75^NTR^ and the activation of TrkA- and p75^NTR^-related phosphoinositide-3-kinase (PI3K)/serine/threonine-protein kinase (AKT) signaling pathways are altered in neurons that are derived from patients suffering from fAD compared to controls. These results provide evidence for altered surface movement of receptors in AD and highlight the importance of investigating receptor dynamics in disease conditions. Uncovering these mechanisms might enable novel therapies for AD.

## 1. Introduction

Changes in lateral diffusion of receptors on the plasma membrane are crucial in determining their functional status. The altered surface movements can result in the formation of membrane protein complexes that activate signaling molecules [1]. For instance, single-molecule tracking experiments showed that the neurotrophin receptors, such as the tropomyosin receptor kinase A receptor (TrkA), also known as high-affinity nerve growth factor receptor, have two distinct motility states on the membrane surface, defined as mobile and immobile phases [1]. The membrane recruitment of downstream intracellular signaling proteins occurs when TrkA is in the immobile phase [1]. It has also been shown that the diffusion coefficient (D) that is calculated from the cell surface movement of receptors provides a good estimate for the effect of a ligand on G protein-coupled receptors [2]. Consequently, the functions of receptors can be predicted by measuring the surface movement parameters of receptor molecules. A large number of diseases are initiated by dysregulation of the receptor-related signaling pathways such as Parkinson’s disease, schizophrenia, and cancer [3,4,5]. Although the expression level and pattern of various receptors have been extensively studied in disease models [6,7] the surface movement of receptors in different diseases has been scarcely investigated. It has recently been published that the surface diffusion of AMPA receptors is disturbed in rodent models of Huntington’s disease and the restoration of the AMPA receptor diffusion rescues memory dysfunction [8]. These data indicate that revealing receptor dynamics in different diseases may contribute to developing new treatments for neurodegenerative diseases such as Alzheimer’s disease (AD).

Alzheimer’s disease is the most common cause of dementia which is characterized by extracellular deposition of beta-amyloid (Aβ) and intracellular neurofibrillary tangles [9]. The familial form of AD (fAD) is caused by genetic mutations, such as mutations in genes for encoding *presenilin (PSEN1* or *PSEN2*) proteins [10]. In AD the unbalanced signaling through the low-affinity binding p75 neurotrophin receptor (p75^NTR^) versus TrkA related signaling [6] plays a pivotal role in changes of altered cellular functions. TrkA and p75^NTR^ activate signaling pathways that are essential to decide the fate of neurons [6,11].

TrkA and p75^NTR^ are frequently co-expressed and interact with each other and they cooperate in mediating signals to survival [12,13]. TrkA alone stimulates pro-survival signaling pathways such as phospholipase C gamma (PLCγ), extracellular signal-regulated kinase 1/2 (ERK1/2), and phosphoinositide-3-kinase (PI3K)/serine/threonine-protein kinases (AKT) pathways [12,14]. On the other hand, in nerve growth factor (NGF) deficiency, TrkA switches from a pro-survival to a pro-apoptotic receptor [15]. p75^NTR^ alone promotes apoptosis via the activation of sphingomyelinase/ceramide or c-jun N-terminal kinase (JNK) pathway but can signal survival when triggering NF-ƙB and PI3/AKT pathways [16,17,18].

Our understanding of AD pathogenesis is currently limited by difficulties in obtaining live neurons from patients, especially in the earlier stages of the disease. Furthermore, the culture of primary human neuronal cells is particularly challenging because of their limited life span [19]. Also, biopsies are often taken from patients at the end-stage of the disease and suitable control tissues that are taken from healthy individuals might be inaccessible due to ethical concerns and potential health risks [20]. Human-induced pluripotent stem cell (hiPSCs) based technology overcomes many of these limitations [21], providing a suitable cellular model for AD. Human iPSCs can be cultured for an unlimited time and differentiated into any type of cell of the human body, such as neurons of the central nervous system (CNS) [22]. We have successfully established and applied AD patient-derived hiPSC lines for modeling AD neuropathology [23,24].

Here, we applied these hiPSCs and performed single-molecule tracking approaches to examine the surface movement of p75^NTR^ and TrkA in living human neurons from *PSEN1* mutant patients and non-demented control subjects. We also examined how downstream signaling pathways, such as ERK1/2, AKT, and JNK1/2 that play essential roles in fAD, could be linked to TrkA and p75^NTR^ activation change in *PSEN1* mutant neurons.

## 2. Results

### 2.1. Characterization of Neuronal Properties of hiPSC-Derived Cultures

hiPSC-derived seven week old neurons were tested to see if they showed neuronal characteristics. All of the cell lines possessed neuron-like morphology and were positively stained for the neuronal markers, β–III tubulin (TUBB3) and microtubule-associated protein 2 (MAP2) (Figure 1A), representing the neuronal differentiation of iPSC derived cell cultures.

The identification of the neuronal phenotype of the hiPSC-derived neurons was conducted by immunolabeling them for glutamatergic (VGLUT1/2), GABAergic (GAD65/67), and cholinergic (VAChT) markers. Our results showed that hiPSC-derived neurons represent a phenotypically mixed population (Figure 1B) in agreement with our earlier published results [24].

In the next step, we examined whether our neuronal differentiation protocol resulted in cells with electrophysiological properties similar to mature neurons. Using the whole-cell patch-clamp technique, the passive electrophysiological parameters of 12 neurons were measured in the cultures. The resting membrane potential and the input resistance of the patched neurons were −64.75 ± 3.9 mV and 742.18 ± 64.59 MΩ, respectively (values are represented as mean ± SEM). Action potentials and voltage-gated Na^+^ and K^+^ currents were also recorded from these cells. As evidence for maturation, the cells exhibited either single or repetitive action potentials in response to positive current injection (Figure 1C). Furthermore, a series of 10 mV depolarization voltage steps from −90 mV resulted in the opening of Na^+^ and K^+^ channels, indicating fast inward and slow outward components, respectively (Figure 1D). Tetrodotoxin (TTX) blocked the inward, while 4-AP blocked the outward currents representing the activation of voltage-gated Na^+^ and K^+^ channels (Figure 1E,F). Overall, the seven week old hiPSC-derived cultures resembled mature neuronal cultures with mixed phenotypes.

### 2.2. Expression and Single-Molecule Detection of TrkA and p75^NTR^ in Control and fAD hiPSC-Derived Neurons

As the balance of TrkA and p75^NTR^ signaling is disturbed in fAD [6], the total TrkA and p75^NTR^ levels in hiPSC-derived neuronal cultures that were obtained from non-demented controls and fAD patients with the *PSEN1* mutation were examined by Western blot. Our results demonstrated that the expression of TrkA significantly decreased in *PSEN1* fAD hiPSC neurons (Figure 2A,B). In contrast, the p75^NTR^ signal was below the detection limit in the cultures of control lines but was abundant in *PSEN1* fAD hiPSC neurons (Figure 2A,B). Consequently, the ratio of p75^NTR^/TrkA was elevated in *PSEN1* mutant neurons (Figure 2C). Next, immunocytochemical stainings for TrkA and p75^NTR^ on fixed hiPSC-derived neuronal cultures that were obtained from non-demented control and *PSEN1* mutant cell lines were performed. The results showed that both neurites and somata expressed TrkA and p75^NTR^ (Figure 2D). In contrast to fixed immunocytochemistry, TrkA and p75^NTR^ molecules were only observed on processes of live cells (Appendix A). To reveal that the moving TrkA and p75^NTR^ molecules are only detected on the neurites, we carried out correlated live-cell single-molecule (TrkA or p75^NTR^) imaging and fixed cell immunocytochemistry (TUBB3) experiments (Appendix A).

### 2.3. Surface Movement Characteristics of TrkA and p75^NTR^ in Control hiPSC Neurons

Live-cell imaging experiments were performed to determine the surface movement parameters of TrkA and p75^NTR^ molecules in the plasma membrane of neurons using TIRF microscopy. After fluorescent labeling of TrkA and p75^NTR^ receptors, 10 s long videos were recorded, single molecule movements were tracked, and then analyzed. Before comparing the diffusion parameters of TrkA and p75^NTR^ in the control and fAD hiPSC neurons, we analyzed the surface dynamics of TrkA and p75^NTR^ in hiPSC neurons that were derived from healthy subjects. Based on the MSD functions, TrkA and p75^NTR^ molecules exhibited two main types of movements on hiPSC neurons that were obtained from non-demented control cell lines: Brownian diffusion, when receptors are moving freely between plasma membrane barriers, and confined motion when the surface movement of receptors are restricted to a small area (Figure 3A,B). On average, p75^NTR^ molecules showed confined movements, while TrkA molecules exhibit Brownian diffusion (Figure 3C). The evaluated average diffusion coefficients of TrkA and p75^NTR^ did not change significantly during the measurement either in the control or *PSEN1* mutant samples (Figure 3D). However, the average diffusion coefficient of TrkA and p75^NTR^ on healthy individual-derived neurites indicated that p75^NTRs^ were moving significantly faster on neurites than TrkAs (Figure 3D; Appendix A).

The diffusion area of p75^NTR^ molecules was more extended on neurites that were derived from healthy individuals than that of TrkA (Figure 3E; Appendix A). Accordingly, the length of p75^NTR^ trajectories was significantly longer (Figure 3F; Appendix A).

To identify the diffusion states of TrkA and p75^NTR^, we used Variational Bayesian-Hidden Markov model (VB-HMM) clustering analysis [25] on the trajectories for each neurite. The analysis showed that the diffusion of both TrkA and p75^NTR^ molecules can be classified into three phases: slow, medium, and fast. Our results suggested that the percentage of the fast fraction increased, while the proportion of slow and medium phases did not change significantly when comparing p75^NTR^ to TrkA molecules (Figure 5C,E; Appendix A). Nevertheless, the average diffusion coefficient of the slow and medium fractions of p75^NTR^ significantly increased compared to TrkA (Figure 5C,E; Appendix A).

### 2.4. Comparison of the Surface Movement of TrkA and p75^NTR^ in Control and PSEN1 fAD hiPSC-Derived Neurons

As surface movement of receptors determine their activation state [26,27], we compared the diffusion parameters of TrkA and p75^NTR^ in healthy individual-derived and *PSEN1* fAD hiPSC neurons. MSD-Δ*t* plots of the trajectories showed that the surface movement of TrkA molecules were less confined in the *PSEN1* fAD hiPSC neurons when compared with the non-demented control samples (Figure 4A). On the other hand, p75^NTR^ molecules exhibited a more confined diffusion mode in the *PSEN1* fAD hiPSC neurons (Figure 4B).

The diffusion area was smaller in the *PSEN1* fAD hiPSC neurons for both TrkA and p75^NTR^ molecules (Figure 4C,D; Appendix A). In addition, the trajectory length that was normalized to steps was also smaller in the case of both TrkA and p75^NTR^ molecules in fAD neurons (Figure 4E,F; Appendix A).

The average diffusion coefficient of TrkA molecules did not change in *PSEN1* fAD hiPSC neurites compared to the non-demented control neurites (Figure 5A; Appendix A). In contrast, the surface movement of p75^NTR^ receptor molecules significantly decreased in the *PSEN1* fAD hiPSC neurites (Figure 5B; Appendix A).

The percentage of slow, medium, and fast fractions of TrkA movement remained unchanged when comparing fAD and control neurites (Figure 5C,D; Appendix A. The average diffusion coefficient of the slow phase decreased, the medium fraction increased, while the fast phase did not change in fAD neurites (Figure 5C,D; Appendix A).

In contrast to TrkA, the proportion of slow trajectories of p75^NTR^ increased while the percentage of fast trajectories decreased in the *PSEN1* fAD hiPSC neurites (Figure 5E,F; Appendix A). The average diffusion coefficient of the slow phase of p75^NTR^ decreased in fAD, but the average diffusion coefficients of the other two fractions remained unchanged in the neurites of *PSEN1* mutant hiPSC neurons (Figure 5E,F; Appendix A).

### 2.5. Expression of the ERK1/2, AKT and JNK1/2 Signaling Pathways in Non-Demented Control and PSEN1 fAD hiPSC-Derived Neurons

Since changes in surface movements of receptors lead to functional changes via the modification of signaling transduction [1], the activation of signaling pathways were examined. To investigate whether TrkA and p75^NTR^-related downstream signaling were altered in *PSEN1* fAD hiPSC neurons, we examined the ERK1/2, AKT, and JNK1/2 signaling pathways. Our Western blot analysis revealed that the expression levels of ERK1/2 and AKT did not change, while the JNK1/2 levels significantly decreased in the cultures of *PSEN1* fAD samples (Figure 6A–C). The activation of ERK1/2 and JNK1/2 pathway was not altered but the AKT signaling pathway was shown to be more active in the *PSEN1* fAD cultures (Figure 6A–C).

## 3. Discussion

Our results showed that the surface trafficking of TrkA and p75^NTR^ are altered in hiPSC-derived neurons that are differentiated from *PSEN1* mutant fAD patients. The surface movement of TrkA molecules was less confined in *PSEN1* mutant neurites. Contrarily, the trafficking of p75^NTR^ molecules was more confined in the fAD neurites. The movement of TrkA and p75^NTR^ both covered smaller areas. The average diffusion coefficient of TrkA did not change, while the average diffusion coefficient of p75^NTR^ decreased in fAD neurites. We revealed that the diffusion state of both TrkA and p75^NTR^ molecules can be classified into three diffusion states: slow, medium, and fast phases. While the percentage of these diffusion states of TrkA was unaffected in the neurons that were derived from *PSEN1* mutant patients, the percentage of the slow component of p75^NTR^ increased and its fast component decreased. In addition, we found that the AKT signaling pathway was more active in *PSEN1* fAD cultures.

We have characterized the neuronal phenotype of the hiPSC-derived cultures. Our results demonstrated that the seven week old cultures expressed neuronal markers such as TUBB3 and MAP2 and showed electrophysiological characteristics of mature neurons. In addition, we tested their neuronal phenotype, revealing that hiPSC-derived neuronal cultures form a mixed population of glutamatergic, GABAergic, and cholinergic neurons when dual-SMAD inhibition-based neuronal differentiation is used [22,24]. We have also validated the fAD patient’s iPSC-based cellular pathology model that was used in this study. Our results showed that *PSEN1* mutant hiPSC neurons produced more Aβ_1-40_ and Aβ_1-42_ and that Aβ_1-42_/Aβ_1-40_ ratio increased compared to the control cells. Previously, we have confirmed that hyperphosphorylation of TAU protein and an increased level of active glycogen synthase kinase 3 beta (GSK3B), a physiological kinase of TAU, is detected in *PSEN1* mutant patient groups [24]. In our previous findings we also demonstrated that γ-secretase inhibitor DAPT treatment on control and *PSEN1* mutant iPSC-derived neurons resulted in reduced endogenous amyloid levels and intracellular accumulation of an AβPP-C-terminal fragment [23].

Next, we have examined the expression pattern of TrkA and p75^NTR^ on the control and *PSEN1* mutant iPSC-derived neurons as TrkA and p75^NTR^ play a critical role in the progress of AD. Although the expression levels of TrkA and p75^NTR^ are differently affected depending on the examined brain area and the progression stage of AD [28], their ratio is essential in determining the functional outcome [6,29]. Our data show that the TrkA levels are decreased, while p75^NTR^ levels are elevated in the *PSEN1* mutant neurites. Consequently, the ratio of p75^NTR^/TrkA is higher in *PSEN1* mutant neurons compared to neurons that were derived from non-demented control cell lines suggesting that the balance is shifted toward the p75^NTR^ activated intracellular signaling in *PSEN1* mutant neurons.

Although the expression pattern of TrkA and p75^NTR^, their interaction, and stimulated signaling pathways leading to AD progression are being extensively studied [6,30], their surface movements in AD have remained unexplored so far. Therefore, in the current study, we investigated the surface trafficking of TrkA and p75^NTR^ on control and *PSEN1* mutant iPSC-derived neurons.

Our data demonstrate that TrkA and p75^NTR^ exhibited different diffusion modes in the control hiPSC-derived cultures. MSD-Δ*t* plots of receptor trajectories indicated that TrkA molecules show Brownian diffusion, while p75^NTR^ molecules display confined motion, and these diffusion modes are altered in *PSEN1* mutant fAD neurons. The lateral diffusion of TrkA molecules was less confined in fAD neurites compared to the non-demented controls, while the diffusion of p75^NTR^ was significantly more confined in fAD neurites. Our finding also shows that the diffusion area of both neurotrophin receptors is smaller in the *PSEN1* mutant neurites. It is now evident that various mechanisms, including the corralling and tethering effect of membrane skeleton [31], lipid complexity [32], lipid-protein interaction [33], and crowding effect of membrane proteins [32,34,35] regulate the dynamics of membrane receptors and ultimately the cell functions. The actin cytoskeleton forms barriers making the membrane compartmentalized. The membrane domains that are generated by the actin fence deviate the dynamics of membrane proteins away from Brownian diffusion [36,37]. Axons and dendrites have a unique membrane cytoskeleton, a periodic cortical actin-spectrin network [38,39,40,41,42] that plausibly also creates special membrane domains causing anomalous diffusion [43]. AD is associated with a disruption of membrane properties such as alterations in membrane lipid composition [44] or increased spectrin proteolysis [45] that may contribute to changes that are seen in the altered diffusion mode and area of TrkA and p75^NTR^ in *PSEN1* mutant neurites.

The surface movements of receptors also indicate their activation state [26,27]. Signal transduction starts with receptor activation, which, in turn, can be precisely described by the changes of their surface trafficking in the plasma membrane [26,27]. Here we show that p75^NTR^ moves faster along the neurites than TrkA in control hiPSCs. In *PSEN1* mutant neurites the average diffusion coefficient of TrkA does not change while p75^NTR^ moves slower. As described earlier, TrkA is immobilized upon ligand binding, which is related to the start of signal transduction [1]; p75^NTR^ has also been shown to slow down and start signalization upon ligand binding [46]. Based on these results our data indicate that TrkA might be less active, while p75^NTR^ might be more active in *PSEN1* mutant neurons as it is also suggested by the increased expression of p75^NTR^/TrkA. The clustering analysis that was based on VB-HMM showed that the percentage of slow, medium, and fast fractions of TrkA did not change, whereas the proportion of slow components of p75^NTR^ molecules increased, their fast fraction decreased, and there was no change in the percentage of the medium fraction. As the average diffusion coefficients of the three fractions did not differ from the values that were calculated for p75^NTR^ in healthy individual-derived neurites, we assumed that the decrease in the average diffusion coefficient of p75^NTR^ in fAD was due to a shift of fast-moving molecules to the slow state.

TrkA and p75^NTR^ are associated with downstream signaling molecules that are important in determining the fate of neurons [6,11,47]. TrkA is generally responsible for signaling survival in response to ligand by inducing ERK1/2 and AKT phosphorylation [12]. On the other hand, p75^NTR^ activation can lead to survival via the AKT signaling pathway [18] but can stimulate apoptotic pathways as well via the JNK pathway [18]. As receptor dynamics determine signaling pathways activation, we examined the changes in the activation of the above-mentioned most relevant TrkA- and p75^NTR^-related signaling pathways in fAD. We found that the AKT pathway is more active in *PSEN1* mutant neurons. This finding is in accordance with a study that performed global transcriptomic analysis of human iPSC lines carrying *PSEN1* mutations and reported that the genes of (PI3K)-AKT signaling pathway are among the most enriched [48].

Even though the activation of the ERK1/2 and JNK pathway did not show any differences between the control and *PSEN1* mutant neurons, we uncovered a significant decrease in the expression levels of JNK and pJNK in fAD. The actual changes of the intracellular downstream actions and finding the connection between the receptor dynamics of TrkA/p75^NTR^ and the intracellular signaling pathways, however, require further investigation and are beyond the scope of this study.

In summary, our data provide evidence that the surface trafficking of TrkA and p75^NTR^ is altered in *PSEN1* mutant hiPSC-derived neurons compared to those of non-demented control hiPSC-derived neurons. These results draw attention to the significance of the investigation of receptor dynamics in disease conditions. Understanding these mechanisms may provide novel therapeutic strategies for the prevention and/or treatment of AD.

## 4. Materials and Methods

### 4.1. Generation of hiPSC-Derived Neurons

A total of three Alzheimer’s disease (AD) patient-derived iPSC lines were used in our experiments. The first one, BIOT-7183-PSEN1 (referred as BIOTi001-A in hPSCreg; https://hpscreg.eu/, accessed on 7 December 2021), bearing a mutation in *PSEN1* gene (*PSEN1* c.265G > C, p.V89L; was already thoroughly characterized (fAD-1; female, age: 55) [49]. The other two iPSC lines were generated from two male siblings bearing the same *PSEN1* c.449T > C, p.L150P mutation (fAD-3 and fAD-4; males, age: 58) as published previously [50]. The patients were clinically diagnosed and characterized by the Institute of Genomic Medicine and Rare Disorders, Semmelweis University, Budapest, Hungary or at the Danish Dementia Research Centre, Rigshospitalet, University of Copenhagen, Denmark, as described previously [24,49,50]. Non-demented volunteers (assessed by clinical evaluation) were used as controls (3 individuals: Ctrl-1, female, age:33; Ctrl-2, female, age:36; Ctrl-5, female, age:56) and all iPSC lines were established, characterized, and maintained under the same conditions, as we published earlier [24]. Neural progenitor cells (NPCs) were generated from each of the hiPSCs by dual inhibition of the SMAD signaling pathway [22] (See details in Appendix A). NPCs then were terminally differentiated into cortical neurons (see details in Appendix A). The extracellular Aβ_1-40_ and Aβ_1-42_ levels were measured in the cultures (Appendix A) and the neuronal differentiation was validated with immunocytochemistry and electrophysiology (see Appendix A for details). Expression of TrkA and p75^NTR^ on hiPSC-derived neurons was identified with immunocytochemistry (Figure 2D, Appendix A).

### 4.2. Single-Molecule Imaging and Analysis of TrkA and p75^NTR^ Molecule Surface Movements in Live Neurons Using Total Internal Reflection Fluorescence (TIRF) Microscopy

Live-cell immunofluorescent labeling was performed to detect TrkA and p75^NTR^ molecules in the plasma membrane of the neurons. Compared to conventional epifluorescence imaging where high background and out-of-focus light is observable, total internal reflection fluorescence (TIRF) microscopy provides a better signal, lower light intensity, and a higher resolution which makes the precise superficial membrane detections and live-cell imaging possible. After the measurement, the diffusion parameters such as the diffusion coefficients; the percentage of slow, medium, and fast fractions; the diffusion area; and the trajectory length of TrkA and p75^NTR^ molecules were calculated. Maximum likelihood estimation [51] was applied to obtain the corresponding diffusion coefficient for each trajectory. Clustering analysis, that was based on a Variational Bayesian for Hidden Markov model, was used to determine the percentage of the slow, medium, and fast fractions of trajectories per neurites [25]. The diffusion area and the length of a molecule trajectory were calculated by a MATLAB script (see Appendix A for details).

### 4.3. Investigation of the Activation of Signaling Pathways in hiPSC-Derived Neuronal Cultures

The activation of TrkA and p75^NTR^-related downstream signaling pathways (ERK1/2, AKT, JNK) were examined in representative samples of a *PSEN1* mutant and a non-demented individual (control) using Western blot (see Appendix A for details).

### 4.4. Statistical Analysis

Values are expressed as mean ± S.E.M. Statistical differences were considered significant at * *p* < 0.05; ** *p* < 0.01; *** *p* < 0.001. All results were analyzed using Statistica 13.3 for Windows (TIBCO) (see details in the Appendix A).

## Figures and Tables

**Figure 1 ijms-22-13260-f001:**
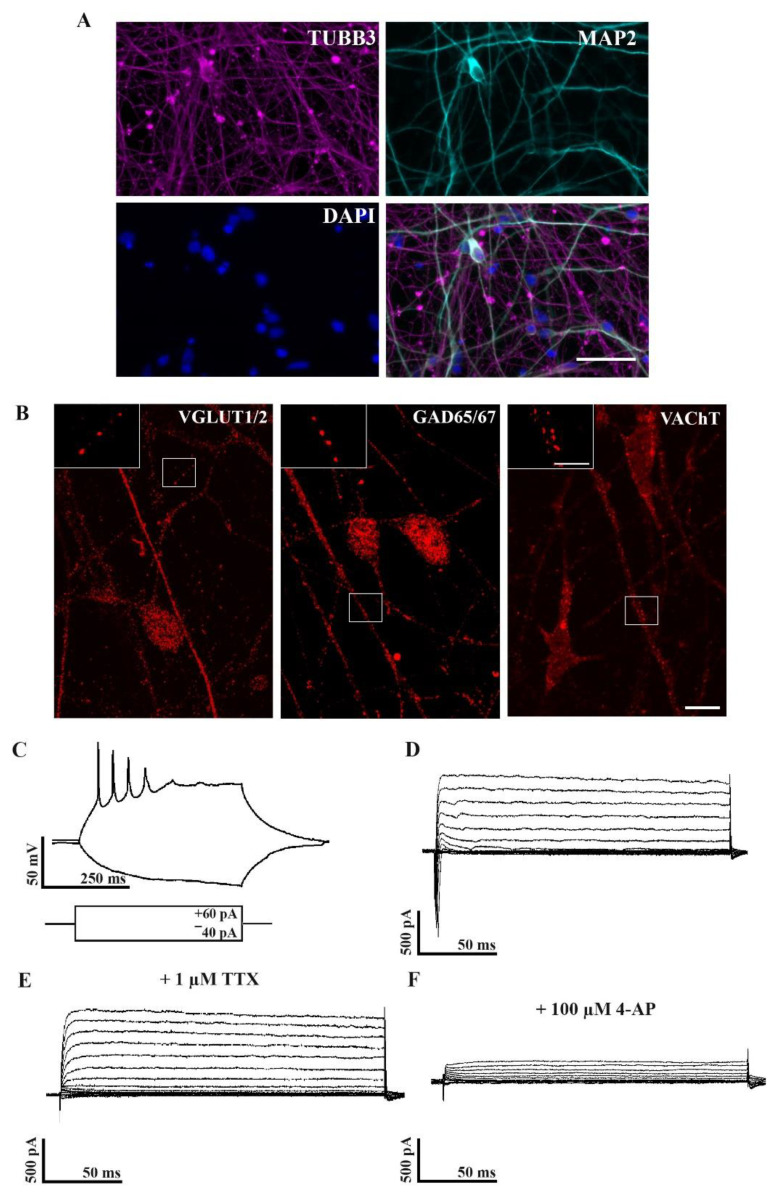
Neuronal phenotypes of differentiated hiPSCs. Immunofluorescence staining of microtubule-associated protein 2 (MAP2, red) and β–III tubulin (TUBB3, green) indicating that the hiPSC-derived neuronal cultures show neuronal phenotypes by week seven (**A**). Nuclei were labeled with DAPI (blue). Scale bar = 50 µm. Images depict VGLUT1/2, GAD65/67, and VAChT immunoreactivity in the cytoplasm and neurites of seven week old human iPSC-derived neurons (**B**). Inserts demonstrate VGLUT1/2, GAD65/67, and VAChT immunoreactive dots on neurites of hiPSCs. IPSC lines: Ctrl-2, fAD-1. Scale bar = 10 µm, insert = 5 µm. Whole-cell patch-clamp recordings show that maturing neurons (seven weeks old) generate action potentials (**C**) and display Na^+^ and K^+^ currents (**D**) that can be blocked by TTX (**E**) or 4-AP (**F**), respectively; *n* = 12.

**Figure 2 ijms-22-13260-f002:**
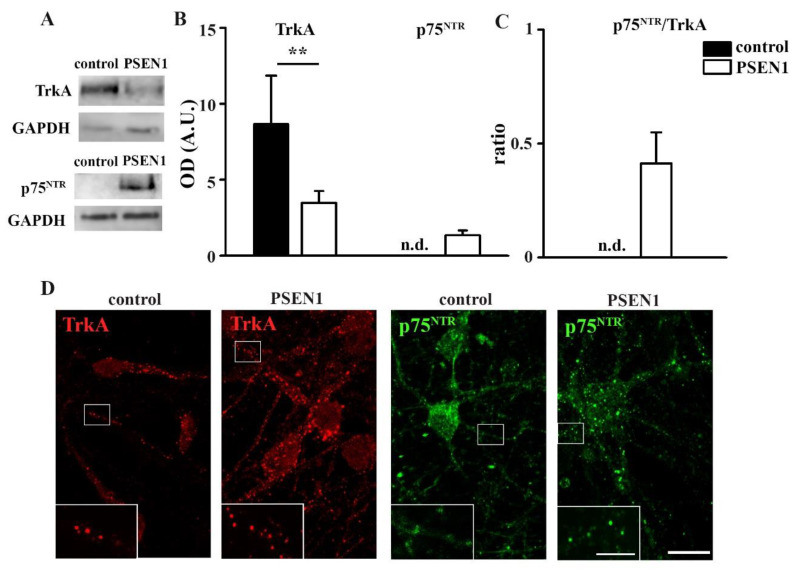
Characterization of TrkA and p75^NTR^ expression in control and *PSEN1* mutant neurons. Representative blots represent TrkA and p75^NTR^ expression in *PSEN1* mutant cultures compared to controls (**A**). IPSC lines: Ctrl-2, fAD-1; *n* = 6. Histograms demonstrate TrkA (**A**), p75^NTR^ (**B**) expression in *PSEN1* mutant neuronal cultures compared to controls and their ratio (**C**) (** *p* < 0.01). Images show TrkA (red) and p75^NTR^ (green) expression in the cytoplasm and neurites of seven week old controls (Ctrl-2) and *PSEN1* mutant neurons (fAD-1) (**D**). The data are expressed in optical densities (pixel/area) ± SEM. Scale bar = 20 µm, scale bar insert = 5 µm.

**Figure 3 ijms-22-13260-f003:**
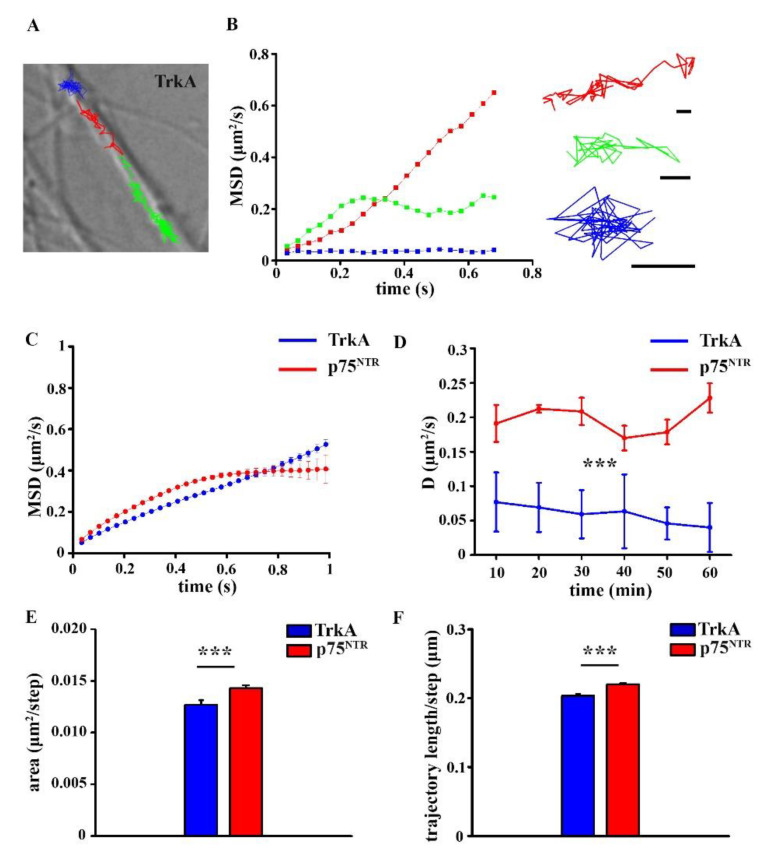
Single-molecule imaging of TrkA and p75^NTR^ in control neurons. Representative trajectories of TrkA molecules on neurites (**A**). Scale bar: 5 µm. The mean square displacement (MSD) functions represent TrkA molecules with different diffusion modes (**B**). Scale bars: 0.5 µm. MSD-Δ*t* plots show the diffusion mode of TrkA and p75^NTR^ in the control neurites (**C**). The diffusion coefficient of TrkA and p75^NTR^ are shown on healthy individual-derived neurites at different time points (**D**). Histograms display the diffusion area (**E**) and the trajectory length (**F**) of TrkA and p75^NTR^ molecules in control neurites. The data are expressed as mean ± SEM (*** *p* < 0.001). The number of trajectories of TrkA in the control and *PSEN1* mutant neurons = 1154; 1246, the number of trajectories of p75^NTR^ in control and *PSEN1* mutant neurons = 1830; 2148. IPSC lines: Ctrl-1, Ctrl-2, Ctrl-3; fAD-1; fAD-2, fAD-3.

**Figure 4 ijms-22-13260-f004:**
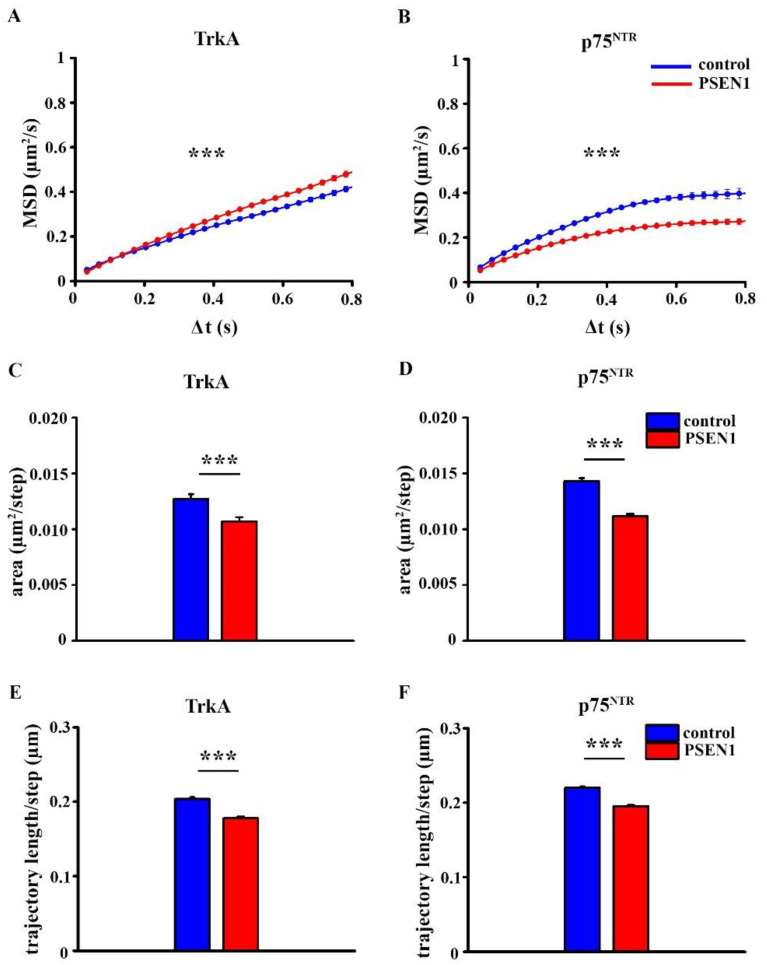
MSD-Δ*t* plots, diffusion area, and trajectory length of TrkA and p75^NTR^ in the control and *PSEN1* mutant neurons. MSD curves show the diffusion modes of TrkA (**A**) and p75^NTR^ (**B**) in non-demented and *PSEN1* mutant neurites. Graphs display the diffusion area of TrkA (**C**) and p75^NTR^ (**D**) in the control and *PSEN1* mutant neurites. Histograms exhibit the trajectory length of TrkA (**E**) and p75^NTR^ (**F**) in control and *PSEN1* mutant neurites. The data are expressed as mean ± SEM (*** *p* < 0.001). The number of trajectories of TrkA in control and *PSEN1* mutant neurons = 1154; 1246, the number of trajectories of p75^NTR^ in control and *PSEN1* mutant neurons = 1830; 2148. IPSC lines: Ctrl-1, Ctrl-2, Ctrl-3; fAD-1; fAD-2, fAD-3.

**Figure 5 ijms-22-13260-f005:**
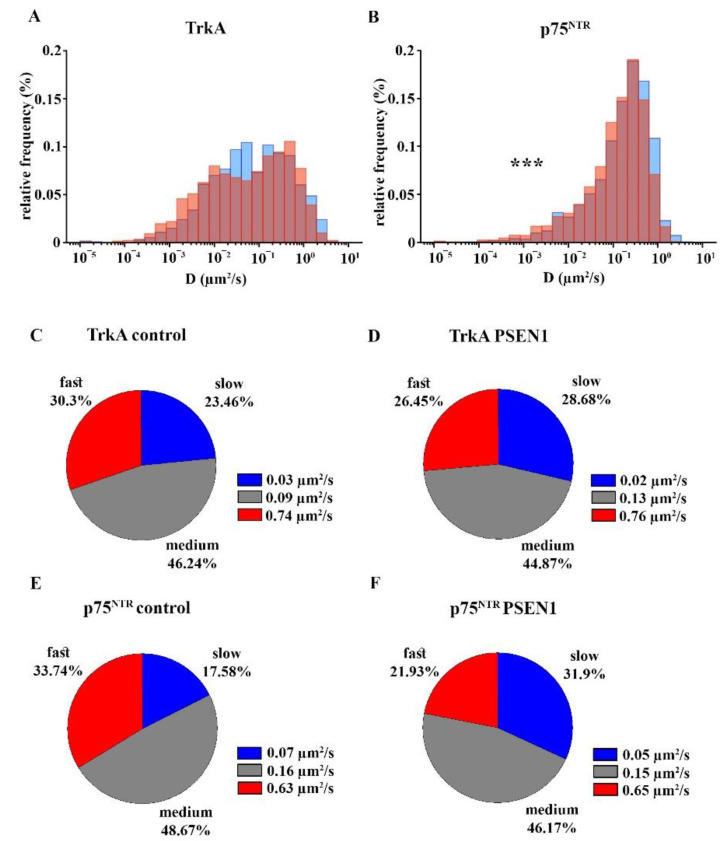
Distribution of the diffusion coefficients and the percentage of slow, medium, and fast fractions of TrkA and p75^NTR^ in the control and *PSEN1* mutant neurons. Logarithmic distribution histograms show the relative frequency of different diffusion intervals in the range of 10^−5^–10 µm^2^/s for TrkA (**A**) and p75^NTR^ (**B**) in control (blue bars) and *PSEN1* mutant neurites (red bars). The number of trajectories of TrkA in control and *PSEN1* mutant neurons = 1154; 1246, p75^NTR^: the number of trajectories of p75^NTR^ in control and *PSEN1* mutant neurons = 1830; 2148. Pie charts demonstrate the percentage of slow, medium, and fast fractions of the trajectories of TrkA (**C**,**D**) and p75^NTR^ (**E**,**F**) per neurite and their average diffusion coefficients in the control (**C**,**E**) and *PSEN1* mutant neurons (**D**,**F**). The data are expressed as mean ± SEM (*** *p* < 0.001). The number of neurites in the control and *PSEN1* mutant cultures for TrkA = 142; 197, the number of neurites in the control and *PSEN1* mutant cultures for p75^NTR^ = 154; 219. IPSC lines: Ctrl-1, Ctrl -2, Ctrl -3; fAD-1; fAD-2, fAD-3.

**Figure 6 ijms-22-13260-f006:**
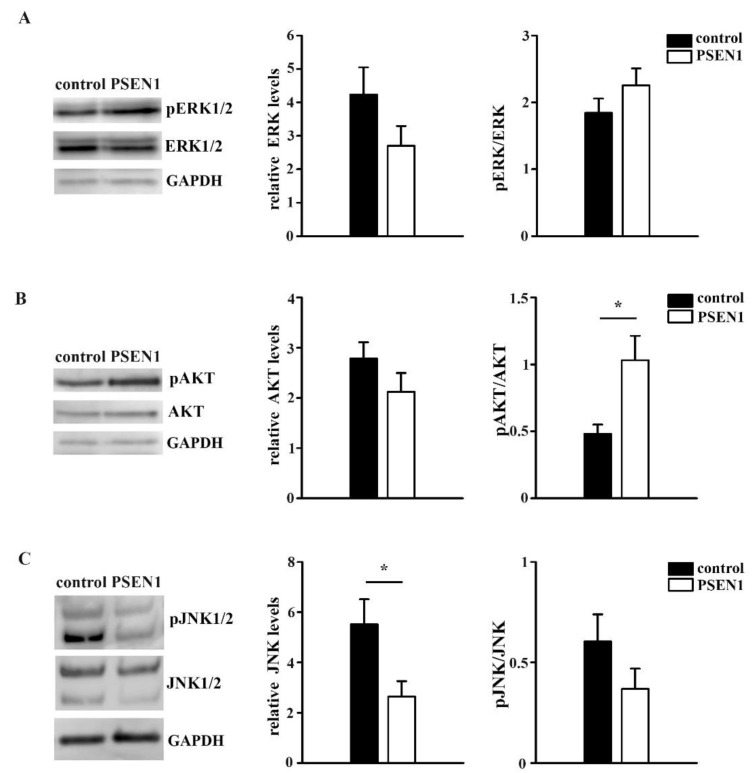
Signaling pathways in the control and *PSEN1* mutant neurons. Representative blots illustrate the expression levels of pERK1/2, ERK1/2, pAKT, AKT, pJNK1/2, and JNK1/2 signaling molecules and GAPDH in *PSEN1* mutant and control cultures. Histograms show the expression of ERK (**A**), AKT (**B**), JNK (**C**) normalized to GAPDH and their ratio of pERK/ERK, pAKT/AKT, and pJNK/JNK in the *PSEN1* mutant cultures compared to controls. The data are expressed in optical densities (pixel/area) ± SEM (* *p* < 0.05), IPSC lines: Ctrl-2, fAD-1; *n* = 6.

## Data Availability

Additional files are made available online along with the manuscript.

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
