# Peer review of "Live-Cell Imaging of Single Neurotrophin Receptor Molecules on Human Neurons in Alzheimer’s Disease"

_ijms, 2021, doi:10.3390/ijms222413260_

Round 1
Reviewer 1 Report
The neurotrophin receptors, including TrkA and the general receptor p75NTR, are essential transmembrane receptor proteins involved in pro-survival signalling critical to neuronal homeostasis, development and survival. Impairments in neurotrophin pathways have been identified in several different neurological disorders, including Azlheimer’s disease (AD). In the manuscript under consideration, the authors have performed live-cell imaging of iPSC-derived neurons from control and PSEN1-AD patients to assess whether TrkA and p75NTR surface dynamics are impacted in the disease. Performing live TIRF microscopy, the authors identified that human PSEN1 neurons display subtle, but significant, differences in neuronal surface movements of both neurotrophin receptors, which may be contributing the some of the observed distinctions in downstream neurotrophin pathways. The manuscript is clearly written and provides useful data to the field. Nevertheless, I have a few concerns, detailed below, which should be addressed before the manuscript is considered for publication. They are as follows:
Major
- It is unclear how the authors know that the live imaging (Supplementary Movies 4-5) is assessing TrkA and p75NTR molecules only on the neuron surface, as opposed to within the cytoplasm. The difference in localisation from the fixed tissue experiments is insufficient to confirm this suggestion. The movements depicted in Movie 4 are very reminiscent of intracellular endosome transport within live neurons – as would be expected if the TrkA/p75NTR were internalised and trafficked. Please provide additional details/characterisation.
Minor
- Red-green colour combinations in immunofluorescent images should be avoided for those who are colour-blind.
- If the authors possess images with counterstains (e.g. with TUBB3), then Fig. 1B and Fig. 2D could be improved to better show the neuronal morphology of the stained cells.
- When discussing the staining of TrkA and p75NTR (Lines 172-175), the authors should highlight whether permeabilisation conditions were used – i.e., were the stainings performed to be able to assess membrane-only proteins? This could explain the discrepancy in localisation with the live imaging. The same goes for the immunohistochemistry performed after the live imaging.
- The authors need to introduce their live-cell imaging experiments in more detail when first highlighted in the Results (Lines 175-178).
- The letter ‘D’ is missing from Figure 2.
- Significance is missing on Fig. 3F – or the text is incorrect (Lines 202-203).
- The pie charts in Figure 5 should be presented in 2D rather than 3D.
- Sample sizes should be provided in all legends.
- Please provide reagent catalogue numbers
- The data presented in Fig. S1 relate to the media of the cells – this should be made clearer in the legend.
Author Response
We would like to thank for constructive comments and time spent on our manuscript. The specific changes are indicated in the original text with track changes and point-by-point answers outlined below.
Answers to Reviewer's comments:
Major
1.It is unclear how the authors know that the live imaging (Supplementary Movies 4-5) is assessing TrkA and p75NTR molecules only on the neuron surface, as opposed to within the cytoplasm. The difference in localisation from the fixed tissue experiments is insufficient to confirm this suggestion. The movements depicted in Movie 4 are very reminiscent of intracellular endosome transport within live neurons – as would be expected if the TrkA/p75NTR were internalised and trafficked. Please provide additional details/characterisation.
RESPONSE: We highly agree that the fixed tissue experiments do not confirm that the detected TrkA and p75NTR molecules are located on the neuron surface in live neurons. As the thickness of plasma membrane is 5 to 10 nm and the angle of incident light in our experiments was set to reach a penetration depth of 100 nm, theoretically we could have visualized the internalized and trafficked receptors as well. Thus, we needed to confirm that the observed receptor molecules by TIRF microscopy are moving in the plasma membrane and not in the cytoplasm. Furthermore, as axonal transport of both TrkA and p75NTR is disturbed in AD [1], it was especially important to ensure that we are imaging receptors on the neuronal surface.
We believe that the imaged receptors are mostly membrane-localized receptors based on two indications. We mainly detected single receptors with our live cell labeling protocol. The fluorescence intensity versus time function indicated one-step photobleaching represented a single ATTO-633 or ATTO-488 fluorophore for TrkA and p75NTR, respectively. The peak intensity of the intensity spot frequency histograms of both TrkA and p75NTR were similar to that of the step sizes for photobleaching. These results suggested that most of the spots represented single fluorophores and single receptors. Since internalized receptors accumulate in endosomes [2,3], and not transported one by one, it is unlikely that we analyzed actively transported receptors.
Additionally, we calculated an estimated speed of TrkA and p75NTR receptors (sum of step distance/time; µm/s). Our results showed that the receptors were moving much faster (TrkA: 1-12 µm/s; p75NTR: 2-10 µm/s) what would have been expected for actively transported receptors (0.1-2.5 µm/s; [4,5].
Taken together, we concluded that most of the TrkA and p75NTR receptors we imaged and analyzed were plasma membrane localized.
Minor
1.Red-green colour combinations in immunofluorescent images should be avoided for those who are colour-blind.
RESPONSE: The red-green colour combination has been changed to magenta-cyan combination.
2.If the authors possess images with counterstains (e.g. with TUBB3), then Fig. 1B and Fig. 2D could be improved to better show the neuronal morphology of the stained cells.
RESPONSE: Unfortunately, we do not possess images with counterstains for these stainings. However, Fig. 1B and Fig. 2D have been improved (contrasted) to make the neuronal morphology more visible.
3.When discussing the staining of TrkA and p75NTR (Lines 172-175), the authors should highlight whether permeabilisation conditions were used – i.e., were the stainings performed to be able to assess membrane-only proteins? This could explain the discrepancy in localisation with the live imaging. The same goes for the immunohistochemistry performed after the live imaging.
RESPONSE: Indeed, we detected TrkA and p75NTR on both soma and neurites on fixed and permeabilized cells, but we only observed the receptors on neurites in non-permeabilized live cells. However, we believe that the difference between live and fix cell labeling is not due to the different staining protocol but the different microscope techniques. The fixed cell labeling was visualized using confocal microscopy, while the live cell labeling was imaged with TIRF microscope. The membrane receptors of the soma of live cells are also likely to be labeled (without permeabilization) but the neurites of hiPSCs are more attached (are closer) to the surface of the coverslip, which is the requirement for TIRF microscopy. TIRF microscope can only detect signals within 200 nm from the surface of the coverslip (the angle of incident light in our experiments was set to reach a penetration depth of 100 nm to optimize the signal/noise ratio). We suppose this is the reason why we could only see signals in neurites in live cell experiments. Our theory is supported by other experiments of our research group carried out on single layered differentiated PC12 cells. These cells are firmly attached to the coverslip; therefore, we could identify AMPA receptors on both soma and neurites [6].
We have made the description of these experiments clearer (Line:178-184):
“The results showed that both neurites and somata express TrkA and p75NTR (Fig. 2. D). In contrast to fixed immunocytochemistry, TrkA and p75NTR molecules were only observed on processes of live cells (Supplementary Movie 4-5). To reveal that the moving TrkA and p75NTR molecules are only detected on neurites, we carried out correlated live-cell single-molecule (TrkA or p75NTR) imaging and fixed cell immunocytochemistry (TUBB3) experiments (Supplementary Movie 3).”
4.The authors need to introduce their live-cell imaging experiments in more detail when first highlighted in the Results (Lines 175-178).
RESPONSE: Thank you for your note. As stated above we have made the description of these experiments clearer (original line: 175-178). In addition, we have introduced the live cell imaging in more detail at the beginning of Section 3.3:
“Live-cell imaging experiments was performed to determine the surface movement parameters of TrkA and p75NTR molecules in the plasma membrane of neurons using TIRF microscopy. After fluorescent labeling of TrkA and p75NTR receptors, 10 seconds long videos were recorded, single molecule movements were tracked and analyzed.”
5.The letter ‘D’ is missing from Figure 2.
RESPONSE: The letter ‘D’ has been added.
6.Significance is missing on Fig. 3F – or the text is incorrect (Lines 202-203).
RESPONSE: Thank you for pointing out this mistake. Data shown on Fig. 3F is statistically significant; it is now indicated on the figure.
7.The pie charts in Figure 5 should be presented in 2D rather than 3D.
RESPONSE: We have changed the presentation of the pie charts; they are presented in 2D.
8.Sample sizes should be provided in all legends.
RESPONSE: Sample sizes are now provided in the Figure Legends.
9.Please provide reagent catalogue numbers
RESPONSE: We have provided the reagent catalogue numbers in the Materials and Methods section.
10.The data presented in Fig. S1 relate to the media of the cells – this should be made clearer in the legend.
RESPONSE: Thank you for your note. It has been corrected.
- Fahnestock, M.; Shekari, A. ProNGF and Neurodegeneration in Alzheimer’s Disease. Front. Neurosci. 2019, 13, 129.
- Marchetti, L.; Callegari, A.; Luin, S.; Signore, G.; Viegi, A.; Beltram, F.; Cattaneo, A. Ligand signature in the membrane dynamics of single TrkA receptor molecules. J. Cell Sci. 2013, 126, 4445–4456, doi:10.1242/jcs.129916.
- Murphy, J.E.; Padilla, B.E.; Hasdemir, B.; Cottrell, G.S.; Bunnett, N.W. Endosomes: A legitimate platform for the signaling train. Proc. Natl. Acad. Sci. 2009, 106, 17615, doi:10.1073/pnas.0906541106.
- Escudero, C.A.; Cabeza, C.; Moya-Alvarado, G.; Maloney, M.T.; Flores, C.M.; Wu, C.; Court, F.A.; Mobley, W.C.; Bronfman, F.C. c-Jun N-terminal kinase (JNK)-dependent internalization and Rab5-dependent endocytic sorting mediate long-distance retrograde neuronal death induced by axonal BDNF-p75 signaling. Sci. Rep. 2019, 9, 6070, doi:10.1038/s41598-019-42420-6.
- Barford, K.; Keeler, A.; McMahon, L.; McDaniel, K.; Yap, C.C.; Deppmann, C.D.; Winckler, B. Transcytosis of TrkA leads to diversification of dendritic signaling endosomes. Sci. Rep. 2018, 8, 4715, doi:10.1038/s41598-018-23036-8.
- Godó, S.; Barabás, K.; Lengyel, F.; Ernszt, D.; Kovács, T.; Kecskés, M.; Varga, C.; Jánosi, T.Z.; Makkai, G.; Kovács, G.; et al. Single-Molecule Imaging Reveals Rapid Estradiol Action on the Surface Movement of AMPA Receptors in Live Neurons. Front. cell Dev. Biol. 2021, 9, 708715, doi:10.3389/fcell.2021.708715.

Reviewer 2 Report
The authors present a tracking study of NGF receptors on the cell surface in fAD (PSEN1) versus control induced neurons. A key technique for this work was total internal reflection fluorescence (TIRF) microscopy. TrkA molecules were less confined in fAD mutant neurites; while, p75NTR molecules were more confined. The average diffusion coefficient for p75NGFR was decreased in fAD. Finally, the proportion of “slow” p75NGFR movement was increased (with “fast” decreasing) in fAD along with elevated AKT signaling. This slowed p75NGFR movement is hypothesized to be a result of more p75NGFR signaling. TrkA signaling is proposed to be reduced. This has significance for the use of induced neurons to model molecular aspects of disease pathogenesis and suggests a role for elevated p75NGFR and reduced TrkA signaling in AD.
Concerns and questions:
- One major issue is that, for figures 2-6 (especially), each figure should give information on the sampling of each of the iPSC lines. Were all three lines used in each fAD and control case? Each figure MUST have sample sizes (n) indicated where it is missing (Fig. 2 and 6).
- It would greatly add to the relevance of the study if the addition or removal of recombinant NGF were shown to affect movement of the receptors in the predicted manner.
- What is the source of “endogenous” NGF in the cultures? Is this physiologically relevant to disease in culture?
Other comments and questions:
- Line 98: what are the identifiers for the control iPSC lines? This information should be provided. Age and sex should also be listed for fAD and control iPSC.
- Line 101: A brief description of the dual inhibition of the SMAD method should be provided here.
- Line 102: A brief description the neural induction procedure should be provided here.
- Methods: how is a “healthy” neurite determined?
- Line 122: Which lines were used for these data?
- Figure 1: Negative control images should be provided to give an indication of background levels of signal.
- Line 203: It is stated, “p75NTR trajectories was significantly longer (Fig. 3. F… ,” however, the figure does not indicate the difference is significant with “*”. Is this statistically significant?
- Lines 216-218: “the percentage of fast fraction increased, while the proportion of slow and medium phases did not change significantly when comparing p75NTR to TrkA molecules (Fig. 5. C, E; Table 218 S4).” This is confusing because the control showed the greatest difference in % slow between TrkA and p75. So slow did change. Was this change not statistically significant?
- Figure 5: For C, D, E, and F, statistical significance should be indicated with stars on the figure where needed.
- Why do the many figure legends indicate a range for the number of trajectories or neurites and not one value?
- Discussion: Is there any reason to believe that the neuronal subtype would affect diffusion characteristics of the receptors?
- Discussion: Is active transport a consideration for these receptors?
Author Response
We would like to thank for constructive comments and time spent on our manuscript. The specific changes are indicated in the original text with track changes and point-by-point answers outlined below.
Answers to Reviewer's comments:
Concerns and questions:
1.One major issue is that, for figures 2-6 (especially), each figure should give information on the sampling of each of the iPSC lines. Were all three lines used in each fAD and control case? Each figure MUST have sample sizes (n) indicated where it is missing (Fig. 2 and 6).
RESPONSE: In case of Figure 1, 2 and 6, representative iPSC lines were used. In single molecule experiments (Figure 3-5), all three control and fAD lines were used. We have now included the information on the sampling and the sample sizes in the Figure Legends.
2.It would greatly add to the relevance of the study if the addition or removal of recombinant NGF were shown to affect movement of the receptors in the predicted manner.
RESPONSE: We highly agree that it would greatly improve the relevance of our findings if we could have determined which factor was responsible for the observed changes in the diffusion parameters of TrkA and p75 NTR in fAD hiPSC-derived neurons.
We performed experiments to find out what might elicit the discovered changes. Since elevated Aβ1-42/Aβ1-40 ratios were detected in the PSEN1 mutant cultures and it is known that Aβ1-42 can activate p75 NTR [1], we first investigated whether Aβ1-42 (synthesized by Dr. Jozsef Kardos, Eötvös Lorand University, [2]) might cause the alterations of diffusion parameters in the PSEN1 mutant neurons. We treated control cells with β-Amyloid (1-42) at different concentrations (100 pM, 100 nM and 1 µM) but the diffusion parameters did not change in the predicted manner.
Next, we examined the role of NGF and proNGF as proNGF\NGF imbalance plays a pivotal role in AD. NGF and proNGF are both ligands of TrkA and p75NTR, although NGF binds to TrkA, while proNGF binds p75NTR with higher affinity [3]. Therefore, we applied NGF2.5S or proNGF on control hiPSC neurons to examine their effects on the diffusion parameters of TrkA and p75NTR. Only one parameter changed the expected way: the diffusion coefficient of p75NTR decreased when control cells were treated with proNGF that can reflect the activation of p75NTR [4]. However, the change did not appear statistically significant probably because of the sample size (number of trajectories=150-250). Further experiments would have been necessary to confirm these findings and our hypothesis that TrkAs are less, while p75NTR are more active in PSEN1 mutant neurons. Unfortunately, in the current series of experiment we could not allocate more resources for the continuation of the experiments, but we plan to go deeper in this direction in the future.
3.What is the source of “endogenous” NGF in the cultures? Is this physiologically relevant to disease in culture?
RESPONSE: Neither the induction and differentiation media nor the culture media contain NGF from external sources, therefore the endogenous source of NGF can only be the neuronal cells themselves. It has been published that neurons produce NGF in the cortex, hippocampus and basal forebrain [5–7]. It has also been established that NGF levels and its signaling are critical in maintaining the balance between cell survival and death in the central nervous system [8], a balance that is disrupted in AD. It is however not verified whether hiPSC-derived neuronal cultures entirely model the disease. For instance, it is not confirmed that NGF levels change the same way in PSEN1 mutant cultures as it occurs during the development of fAD. Consequently, we do not know if NGF production in PSEN1 mutant cultures is physiologically relevant to AD but the possibility can not be excluded.
Other comments and questions:
1.Line 98: what are the identifiers for the control iPSC lines? This information should be provided. Age and sex should also be listed for fAD and control iPSC.
RESPONSE: In our paper, we are referring to our previous papers, among a ‘Lab Resource’-type methodical paper that describes the establishment of the human iPSC lines. However, we agree with the reviewer, this information should be mentioned here as well for all cell lines, therefore we supplemented the material and methods section as follows:
Line: 115-116:
“Three Alzheimer’s disease (AD) patient-derived iPSC lines were used in our experiments. The first one, BIOT-7183-PSEN1 (referred as BIOTi001‑A in hPSCreg; https://hpscreg.eu/), bearing a mutation in PSEN1 gene (PSEN1 c.265G>C, p.V89L; was already thoroughly characterized (fAD-1; female, age: 55) [25]. The other two iPSC lines were generated from two male siblings, bearing the same PSEN1 c.449T>C, p.L150P mutation (fAD-3 and fAD-4; males, age: 58) as published previously [26].”
Line: 120-122:
“Non-demented volunteers (assessed by clinical evaluation) were used as controls (3 individuals) and all iPSC lines (Ctrl-1, female, age:33; Ctrl-2, female, age:36; Ctrl-5, female, age:56;) were established, characterized, and maintained under the same conditions, as we published earlier [24].”
2.Line 101: A brief description of the dual inhibition of the SMAD method should be provided here.
RESPONSE: Indeed, this information is provided in detail in the Supplementary Material as follows:
“Neural progenitor cells (NPCs) were generated from each of the hiPSCs by dual inhibition of the SMAD signaling pathway using LDN193189 and SB431542 [9]. Neural induction was initiated upon reaching approx. 90% confluence of iPSCs on Matrigel-coated dishes by addition of Neural Induction Medium (NIM) (1:1 (v/v) mixture of Dulbecco’s Modified Eagle’s/F12 and Neurobasal Medium, 1x N-2 Supplement, 1x B-27 Supplement, 1x Nonessential Amino Acids (NEAA), 2 mM L-Glutamine, 50 U/ml Penicillin/Streptomycin, 100 μM β-mercaptoethanol, 5 μg/ml insulin), which was supplemented with 5 ng/ml basic fibroblast growth factor (bFGF), 0.2 µM LDN193189 (Selleckchem) and 10 μM SB431542. The NIM medium was changed every day. On day 10, neural rosettes were picked manually and re-plated on poly-L-ornithine/laminin (POL/L; 0.003%/3 µg/cm2) coated dishes and expanded in Neural Maintenance Medium (NMM) (1:1 (v/v) mixture of Dulbecco’s Modified Eagle’s/F12 and Neurobasal Medium, 1x N-2 Supplement, 1x B-27 Supplement, 1x NEAA, 2 mM L-Glutamine, 50 U/ml Penicillin/Streptomycin), and supplemented with 10 ng/ml epidermal growth factor (EGF) and 10 ng/ml bFGF.”
We hope that this section contains all the information which describes the procedure.
3.Line 102: A brief description the neural induction procedure should be provided here.
RESPONSE: The neuronal induction procedure is described in detail in the Supplementary Materials. Upon your comment, we supplemented the main text and referred to the Supplementary Material more clearly.
Line:124-126:
“Neural progenitor cells (NPCs) were generated from each of the hiPSCs by dual inhibition of the SMAD signaling pathway [22] (See details in Supplementary Material). NPCs then were terminally differentiated into cortical neurons (See details in Supplementary Material). The extracellular Aβ1-40 and Aβ1-42 levels were measured in the cultures and the neuronal differentiation was validated with immunocytochemistry and electrophysiology (Please see Supplementary Material file for details)”.
4.Methods: how is a “healthy” neurite determined?
RESPONSE: Thank you for your note. Here we mean derived from a healthy individual = non-dement individual. This wording might be misleading, therefore we changed the wording and stated: “healthy individual-derived” or “control”.
5.Line 122: Which lines were used for these data?
RESPONSE: Ctrl-2 and fAD-1 lines were used for the Western blot experiments. We have now indicated in the Figure Legends of Figure 6 which lines were used in the above experiments.
6.Figure 1: Negative control images should be provided to give an indication of background levels of signal.
RESPONSE: We have provided a negative control image for Figure 1. Please find it in the Supplementary materials (Fig. S2.)
7.Line 203: It is stated, “p75NTR trajectories was significantly longer (Fig. 3. F… ,” however, the figure does not indicate the difference is significant with “*”. Is this statistically significant?
RESPONSE: Thank you for pointing out this mistake. It is statistically significant; it is now indicated on the figure.
8.Lines 216-218: “the percentage of fast fraction increased, while the proportion of slow and medium phases did not change significantly when comparing p75NTR to TrkA molecules (Fig. 5. C, E; Table 218 S4).” This is confusing because the control showed the greatest difference in % slow between TrkA and p75. So slow did change. Was this change not statistically significant?
RESPONSE: We compared the percentage of slow, medium, and fast fractions of TrkA and p75NTR molecules with Mann-Whitney U test, which gave the result that only the percentage of fast fractions changed significantly.
9.Figure 5: For C, D, E, and F, statistical significance should be indicated with stars on the figure where needed.
RESPONSE: Figure 5 C, D, E, F compares the percentage of slow, medium, and fast phases between TrkA control and p75NTR control, TrkA control and TrkA PSEN1, p75NTR control and p75NTR PSEN1. In addition, it compares the diffusion coefficients of slow, medium, and fast phases between the TrkA control and p75NTR control, TrkA control and TrkA PSEN1, p75NTR control and p75NTR PSEN1. We tried to indicate the significance with stars on the pie charts, but the figure became too complicated. So, although we agree that it would be better see the significance on the figure, we could only demonstrate significances in the text.
10.Why do the many figure legends indicate a range for the number of trajectories or neurites and not one value?
RESPONSE: The numbers found in the figure legends are values. However, the way they were displayed was confusing, so we have changed it in the text: e.g.: “the number of trajectories of TrkA=1154-1246” has been changed to “number of trajectories of TrkA in control and PSEN1 mutant neurons=1154; 1246.”
11.Discussion: Is there any reason to believe that the neuronal subtype would affect diffusion characteristics of the receptors?
RESPONSE: Although it is possible that neuronal subtypes affect diffusion characteristics, we could not find any data in the scientific literature supporting this perception. Furthermore, both the control and the PSEN1 mutant cultures represented a phenotypically mixed population, and we did random sampling, which diminishes any potential differences.
12.Discussion: Is active transport a consideration for these receptors?
RESPONSE: As active transport of both TrkA and p75NTR is disturbed in AD [8], it was important to examine whether we are imaging receptors on the neuronal surface or internalized and trafficked receptors. Furthermore, as the thickness of plasma membrane is 5 to 10 nm and the angle of incident light in our experiments was set to reach a penetration depth of 100 nm, theoretically we could have visualized the internalized and trafficked receptors as well. Thus, we needed to confirm that the observed receptor molecules by TIRF microscopy are moving in the plasma membrane and not in the cytoplasm.
We believe that the imaged receptors are mostly membrane-localized receptors based on two indications. We mainly detected single receptors with our live cell labeling protocol. The fluorescence intensity versus time function indicated one-step photobleaching represented a single ATTO-633 or ATTO-488 fluorophore for TrkA and p75NTR, respectively. The peak intensity of the intensity spot frequency histograms of both TrkA and p75NTR were similar to that of the step sizes for photobleaching (Supplementary Fig.3.). These results suggested that most of the spots represented single fluorophores and single receptors. Since internalized receptors accumulate in endosomes [3,10], and not transported one by one, it is unlikely that we analyzed actively transported receptors.
Additionally, we calculated an estimated speed of TrkA and p75NTR receptors (sum of step distances/time; µm/s). Our results showed that the receptors were moving much faster (TrkA: 1-12 µm/s; p75NTR: 2-10 µm/s) what would have been expected for actively transported receptors (0.1-2.5 µm/s; [11,12].
Taken together, we concluded that most of the TrkA and p75NTR receptors we imaged and analyzed were plasma membrane localized.
- Yaar, M.; Zhai, S.; Pilch, P.F.; Doyle, S.M.; Eisenhauer, P.B.; Fine, R.E.; Gilchrest, B.A. Binding of beta-amyloid to the p75 neurotrophin receptor induces apoptosis. A possible mechanism for Alzheimer’s disease. J. Clin. Invest. 1997, 100, 2333–2340, doi:10.1172/JCI119772.
- Szegő, É.M.; Csorba, A.; Janáky, T.; Kékesi, K.A.; Ábrahám, I.M.; Mórotz, G.M.; Penke, B.; Palkovits, M.; Murvai, Ü.; Kellermayer, M.S.Z.; et al. Effects of Estrogen on Beta-Amyloid-Induced Cholinergic Cell Death in the Nucleus Basalis Magnocellularis. Neuroendocrinology 2011, 93, 90–105, doi:10.1159/000321119.
- Marchetti, L.; Callegari, A.; Luin, S.; Signore, G.; Viegi, A.; Beltram, F.; Cattaneo, A. Ligand signature in the membrane dynamics of single TrkA receptor molecules. J. Cell Sci. 2013, 126, 4445–4456, doi:10.1242/jcs.129916.
- Marchetti, L.; Bonsignore, F.; Gobbo, F.; Amodeo, R.; Calvello, M.; Jacob, A.; Signore, G.; Schirripa Spagnolo, C.; Porciani, D.; Mainardi, M.; et al. Fast-diffusing p75(NTR) monomers support apoptosis and growth cone collapse by neurotrophin ligands. Proc. Natl. Acad. Sci. U. S. A. 2019, 116, 21563–21572, doi:10.1073/pnas.1902790116.
- Mufson, E.J.; Conner, J.M.; Varon, S.; Kordower, J.H. Nerve growth factor-like immunoreactive profiles in the primate basal forebrain and hippocampal formation. J. Comp. Neurol. 1994, 341, 507–519, doi:https://doi.org/10.1002/cne.903410407.
- Hayashi, M.; Yamashita, A.; Shimizu, K.; Sogawa, K.; Fujii, Y. Expression of the gene for nerve growth factor (NGF) in the monkey central nervous system. Brain Res. 1993, 618, 142–148, doi:https://doi.org/10.1016/0006-8993(93)90437-R.
- Biane, J.; Conner, J.M.; Tuszynski, M.H. Nerve growth factor is primarily produced by GABAergic neurons of the adult rat cortex. Front. Cell. Neurosci. 2014, 8, doi:10.3389/fncel.2014.00220.
- Fahnestock, M.; Shekari, A. ProNGF and Neurodegeneration in Alzheimer’s Disease. Front. Neurosci. 2019, 13, 129.
- Chambers, S.M.; Fasano, C.A.; Papapetrou, E.P.; Tomishima, M.; Sadelain, M.; Studer, L. Highly efficient neural conversion of human ES and iPS cells by dual inhibition of SMAD signaling. Nat. Biotechnol. 2009, 27, 275–280, doi:10.1038/nbt.1529.
- Murphy, J.E.; Padilla, B.E.; Hasdemir, B.; Cottrell, G.S.; Bunnett, N.W. Endosomes: A legitimate platform for the signaling train. Proc. Natl. Acad. Sci. 2009, 106, 17615, doi:10.1073/pnas.0906541106.
- Escudero, C.A.; Cabeza, C.; Moya-Alvarado, G.; Maloney, M.T.; Flores, C.M.; Wu, C.; Court, F.A.; Mobley, W.C.; Bronfman, F.C. c-Jun N-terminal kinase (JNK)-dependent internalization and Rab5-dependent endocytic sorting mediate long-distance retrograde neuronal death induced by axonal BDNF-p75 signaling. Sci. Rep. 2019, 9, 6070, doi:10.1038/s41598-019-42420-6.
- Barford, K.; Keeler, A.; McMahon, L.; McDaniel, K.; Yap, C.C.; Deppmann, C.D.; Winckler, B. Transcytosis of TrkA leads to diversification of dendritic signaling endosomes. Sci. Rep. 2018, 8, 4715, doi:10.1038/s41598-018-23036-8.
Round 2
Reviewer 1 Report
The authors have responded constructively to all comments and the manuscript is improved. However, before acceptance, the suggested changes to the figures should be implemented - e.g. the pie charts are still in 3D and there are still red/green immunofluorescent images.
Author Response
Dear Reviewer,
Thank you for your comments that helped to improve our manuscript. We appreciate the time and effort that you have dedicated to providing your valuable feedback.
We have reuploaded the revised figures with the suggested changes. We will highlight the modifications to the journal and ask the editor to change the figures to the modified version.
Reviewer 2 Report
Concerns have been sufficiently addressed by the edits provided by the authors.
Author Response
Dear Reviewer,
Thank you for your comments that helped to improve our manuscript. We appreciate the time and effort that you have dedicated to providing your valuable feedback.